# Synthetic Biology Meets Ca^2+^ Release-Activated Ca^2+^ Channel-Dependent Immunomodulation

**DOI:** 10.3390/cells13060468

**Published:** 2024-03-07

**Authors:** Bernadett Bacsa, Valentina Hopl, Isabella Derler

**Affiliations:** 1Division of Medical Physics und Biophysics, Medical University of Graz, A-8010 Graz, Austria; bernadett.bacsa@medunigraz.at; 2Institute of Biophysics, JKU Life Science Center, Johannes Kepler University Linz, A-4020 Linz, Austria; valentina.hopl@jku.at

**Keywords:** CRAC channels, STIM1, Orai1, calcium (Ca^2+^) synthetic biology, chemical inducers of dimerization, proteolytic cleavage, optogenetics, engineered immune cells

## Abstract

Many essential biological processes are triggered by the proximity of molecules. Meanwhile, diverse approaches in synthetic biology, such as new biological parts or engineered cells, have opened up avenues to precisely control the proximity of molecules and eventually downstream signaling processes. This also applies to a main Ca^2+^ entry pathway into the cell, the so-called Ca^2+^ release-activated Ca^2+^ (CRAC) channel. CRAC channels are among other channels are essential in the immune response and are activated by receptor–ligand binding at the cell membrane. The latter initiates a signaling cascade within the cell, which finally triggers the coupling of the two key molecular components of the CRAC channel, namely the stromal interaction molecule, STIM, in the ER membrane and the plasma membrane Ca^2+^ ion channel, Orai. Ca^2+^ entry, established via STIM/Orai coupling, is essential for various immune cell functions, including cytokine release, proliferation, and cytotoxicity. In this review, we summarize the tools of synthetic biology that have been used so far to achieve precise control over the CRAC channel pathway and thus over downstream signaling events related to the immune response.

## 1. Introduction—The Role of Ca^2+^ Release-Activated Ca^2+^ (CRAC) Channels in Immunology

Calcium ions (Ca^2+^) are the most widely used intracellular messengers that actuate numerous biological processes within the human body [1,2,3,4]. A major Ca^2+^ entry pathway is activated by the depletion of intracellular Ca^2+^ stores and is thus called the store-operated Ca^2+^ entry (SOCE) route. A key player in SOCE is the Ca^2+^ release-activated Ca^2+^ (CRAC) channel [5,6,7,8,9], which plays a versatile role in numerous biological processes, including but not restricted to immune system function [10,11,12,13,14,15]. In immune defense, these channels play a decisive role in the innate and adaptive immune response [10,14,16]. In the innate immune system, as a fast and non-specific defense system that reacts to any germs entering the human body, CRAC channels contribute to the function of neutrophils, macrophages, monocytes, dendritic cells (DCs), natural killer (NK) cells, and mast cells [10,16]. If the innate immune response is unable to destroy the invading germs, the adaptive immune system, consisting of T cells, B cells, and antibodies, takes over the task, which is also subject to the control of CRAC channels [10,11,12,13,14,15]. Dysregulations in CRAC channel function can lead to severe diseases [11,15,16,17,18,19,20,21,22] such immunodeficiencies or cancer.

In a first step, CRAC entry into lymphocytes is triggered via receptor stimulation at the cell membrane; for instance, by binding of an antigen to the T or B cell receptor (TCR or BCR), or by the formation of antigen–antibody complexes to the Fc-receptor for IgE (FcεRII) on mast cells or to Fc-receptors for IgG (FcγRs) on dendritic cells (DCs), natural killer (NK) cells, or macrophages [10,14,15,23,24,25,26,27]. This initiates a cascade of protein phosphorylation, leading to the activation of phospholipase Cγ (PLCγ) and the production of second messengers. PLCγ hydrolyzes phosphatidylinositol-4,5-bisphosphate (PI(4,5)P_2_) into the soluble head group inositol-1,4,5-trisphosphate (IP_3_) and the PM-associated lipid diacylglycerol (DAG). Diffusible IP_3_ binds to IP_3_R and triggers transient Ca^2+^ release from the ER [5,9]. The reduction in Ca^2+^ concentration in the ER is perceived by the stromal interaction molecules (STIM) 1 and 2 in the ER membrane [28,29,30]. STIM proteins translocate to the ER-PM junctions to bind to and open the CRAC channel pore-forming complex Orai in the PM, formed by Orai1 and its homologues Orai2 and Orai3 [9,31,32,33,34,35]. The resulting influx of extracellular Ca^2+^ causes sustained elevation of Ca^2+^ levels, which is required for the immune response [14,15,36,37,38,39]. In lymphocytes in particular, Orai1 has been considered as essential, as patients with mutations in Orai1 display a SCID-like immunodeficiency [38]. Nevertheless, the other two isoforms, Orai2 and Orai3, also form part of the CRAC channel in immune cells [40]. Among the STIM isoforms, both STIM1 and STIM2 play essential roles in lymphocytes [10,11,12,13,14,16], although STIM1 has been suggested to be more dominant [10,14,41].

During the innate immune response, Ca^2+^ mobilization results in the recruitment and activation of neutrophils, monocytes, macrophages, and dendritic cells [16,42,43]. Among the different phagocytotic innate immune cells, the roles of STIM and Orai proteins are best characterized in neutrophil function, especially in phagocytosis, degranulation, cytokine release, and radical oxygen species (ROS) production [44,45,46,47,48,49]. CRAC-dependent Ca^2+^ entry promotes the activation and maturation of dendritic cells while also playing a role in their phagocytic activity [50,51,52,53]. In addition, macrophages require SOCE for the production of tumor necrosis factor (TNF-α) [48] and nitric oxide [54,55], as well as phagocytosis. However, due to limited data, the exact role of CRAC channels in monocyte or macrophage function is still unclear [16,56].

As unique effectors of innate responses, mast cells are the major drivers of IgE-mediated allergic responses, including allergies, atopic dermatitis, asthma, and anaphylaxis. CRAC channel activity contributes to mast cell activation via FcεRI receptors, which is an important driver for the secretion and synthesis of allergic mediators including histamine, proteases, prostaglandins, and leukotrienes [57,58,59,60,61,62,63]. Therefore, Ca^2+^ influx via CRAC channels controls the degranulation of cytosolic vesicles and the release of proinflammatory cytokines, as demonstrated in mast cells, natural killer (NK) cells, and cytotoxic T cells (CTL) [58,60,61,63,64].

In adaptive lymphocytes, several lines of evidence suggest that Ca^2+^ signaling plays an important role in the development and positive and negative selection of T and B lymphocytes [14,65,66,67,68,69,70,71]. Furthermore, Ca^2+^ signaling in T and B cells is responsible for short-term and long-term immune responses [72]. Short-term functions include the activation of lymphocytes, which is ensured by the stable establishment of the immunological synapse [73,74,75,76] and is associated with reduced lymphocyte motility [27,71,77,78]. Moreover, secretion and cell death of the target cell occur within seconds [13,64,79,80,81,82,83,84]. Long-term functions, which are only established when the immunological synapse and the enhanced Ca^2+^ levels persist for hours, represent Ca^2+^-dependent gene expression, which determines effector functions and differentiation [39,85,86,87,88]. The duration and strength of the Ca^2+^ signal defines the pattern of gene expression [14,15,39,89,90,91]. Gene expression can be controlled by numerous Ca^2+^-dependent enzymes, including calcineurin, calmodulin kinases, and transcription factors such as NFAT [39,92,93,94], NF-kB [95,96], and CREB [97]. Therefore, SOCE is essential for the regulation of chemokine and cytokine gene expression, including interleukins (IL-2, IL-4, IL-10, IL-17), interferon-γ (IFNγ), and tumor necrosis factor (TNF). It also regulates several metabolic pathways such as glycolysis and mitochondrial respiration, thereby controlling lymphocyte proliferation and effector functions [14,88,89,90,91,98].

In addition to maintaining a healthy immune response, a close relationship between CRAC channels and disease through impaired store-operated Ca^2+^ entry (SOCE) in patients with immunodeficiencies became apparent 15 years before the molecular nature of CRAC channels was identified [31,38,84,99,100]. RNAi screening and genetic linkage analysis in patients with inherited defects in CRAC channel function eventually led to the discovery of STIM1 and Orai1 in 2005/2006 [28,29,34,35,38]. Meanwhile, a series of homozygous loss- and gain-of-function (LoF and GoF) mutations in Orai1 and STIM1 have been identified to cause CRAC channelopathies [30,38,101,102,103,104,105,106,107,108,109,110,111,112,113,114,115] due to an imbalance in the cellular Ca^2+^ homeostasis. Severe combined immunodeficiency (SCID) with recurrent and chronic viral, bacterial, and fungal infections occurs in patients who completely lack SOCE due to an inherited LoF mutation in the Orai1 and Stim1 genes. Additionally, these patients with immunodeficiencies suffer from humoral autoimmunity, characterized by the presence of autoantibodies targeting erythrocytes and platelets, leading to the development of hemolytic anemia and thrombocytopenia [30,107,116,117]. Furthermore, they can develop a variety of severe non-immunological symptoms. GoF mutations in these CRAC channel components have been shown to lead to dysfunctions such as York platelet and Stormorken-like syndrome and tubular aggregate myopathy, autoimmunity, muscular hypotonia, and ectodermal dysplasia [21].

In addition to the CRAC channels’ role in many immune cell functions, they are also potential drug targets for the treatment of various inflammatory reactions, allergic diseases [118,119], and many cancer types [20,120,121,122,123,124,125,126,127,128]. Several studies have confirmed that the genetic deletion or inhibition of CRAC channels hinders T cell and mast cell functions and diminishes inflammation of autoimmunity, transplant rejection, and asthma [119]. Particularly, the development and the progression of several T cell-mediated autoimmune diseases, including inflammatory bowel disease [129] (IBD), experimental autoimmune encephalomyelitis [130,131] (EAE), graft-versus-host disease [132,133] (GvHD), and psoriasis form skin inflammation [134,135,136], have been linked to Orai and STIM proteins. Furthermore, CRAC channels have recently been found to play a role in the aging-related deterioration of the immune system, namely immunosenescence, as altered expression of CRAC channel components affects Ca^2+^ homoeostasis in specific T cell subsets [137,138].

The potential role of CRAC channels in cancer development and progression has been deeply discussed in several recent review articles [120,122,123,126,128,139,140,141,142,143,144]. However, the importance of CRAC channels has been also revealed in Ca^2+^-dependent tumor killing by a subset of lymphocytes such as CTL and NK cells. They eliminate cancer cells by releasing cytotoxic or lytic granules containing perforin and granzymes at the immune synapse between cytotoxic cells and cancer cells [63,82]. Since CRAC channels are upregulated in many cancers, partial downregulation or inhibition of one of its components in CTLs could increase perforin-dependent cancer killing and simultaneously impede tumor growth within the tumor microenvironment [64,145,146].

To summarize, the CRAC channel components STIM and Orai play an essential role in the immune response. It is therefore not surprising that their dysregulation can lead to severe diseases, thus making them important targets for therapeutic intervention.

## 2. CRAC Channel Working Mechanisms

The CRAC channel forms a complex of STIM [28,29,31] and Orai [33,34,35] isoforms. In general, the presence of STIM1 and Orai1 is sufficient to fully reconstitute CRAC channel function. However, their function can be modulated by STIM2, Orai2, and Orai3, resulting in a variety of CRAC channels of different compositions. This ensures cell type-specific regulation by CRAC channels and is relevant in the development of diseases [121,122,140,147,148,149,150,151,152,153,154,155,156]. Their activation mechanisms have been summarized in detail in previous reviews [9,32,154,157,158,159,160,161,162,163,164,165]. In the following paragraphs, only the most important activation steps and mechanisms necessary for understanding the following chapters on the application of synthetic biology tools to the CRAC components STIM1 and Orai1 are briefly described. As synthetic biology tools have not yet been applied to other STIM and Orai isoforms, the differences between the properties and mechanisms of the different isoforms are beyond the scope of this review, but they are described in detail elsewhere [166].

STIM1 senses the ER-Ca^2+^ concentration via its EF-sterile alpha motif (EF-SAM) pointing into this organelle. In the resting state, Ca^2+^ binds to the EF-SAM domain, which stabilizes STIM1 in the quiescent state [32,167,168,169,170]. Upon Ca^2+^ store depletion, the STIM1 Ca^2+^ sensor loses the bound Ca^2+^, which serves as a trigger for structural rearrangements within the entire STIM1 protein to assume an active conformation [171,172,173,174,175,176]. These structural changes include an unfolding of the N-terminal segment, which is conferred via the short transmembrane domain to the cytosolic region of STIM1 [32,163,177,178]. The STIM1 C-terminus is tightly folded in the resting state [174,175], while it elongates upon store depletion [172,175,176,178]. It includes three coiled-coil (CC) regions, which contribute to the maintenance of the quiescent state [104,172,174,175,176] and are involved in the coupling to the counterpart Orai1 in the elongated conformation [179,180,181,182]. Several small STIM1 C-terminal fragments have been identified to be sufficient for Orai1 activation, including the CRAC activation domain (CAD, aa 342-448), the STIM1 Orai activating region (SOAR, aa 344-442), the coiled-coil domain-containing region b9 (CCb9, aa 339-444), or the Orai-activating small fragment (OASF, aa 233-474) [183,184,185,186]. Following the coiled-coil segment, the STIM1 C-terminus additionally contains a flexible region containing a CRAC modulatory domain [187] as well as lipid binding domains [188,189,190].

Following store depletion, structural remodeling of the STIM1 C-terminus allows it to bridge the distance between the ER and PM at junctions where the membranes are only 10–25 nm apart from each other [191]. There, STIM1 forms tight contacts with the Orai1 channel, which forms a hexameric complex [192,193,194,195] with each subunit composed of four TM domains, a cytosolic N- and C-terminus, and two extracellular and one intracellular loop [33,35,38]. The Ca^2+^ ion pore is formed in the center by six TM1 domains, which is surrounded by TM2 and TM3 and at the complex periphery by TM4 [192,193,194,195]. TM4 is connected via a bent region, the so-called nexus [196], to the C-terminus, with the latter functioning as the main coupling site for STIM1 [179]. It is currently assumed that STIM1 coupling to the Orai1 C-terminus induces a signal that leads to a global conformational change of the entire channel, which results in the opening of the pore [159,166,197,198,199,200]. The cytosolic loop2 has been reported to contribute to Orai1 gating via direct coupling to STIM1 [201,202,203]. Also, the Orai1 N-terminus is involved in STIM1-mediated Orai1 activation [202,204,205,206]. However, the extent to which the N-terminus and loop2 contribute to STIM1-mediated activation and the molecular nature of their potential binding pockets remains unknown. In summary, Ca^2+^ store depletion-induced CRAC channel activation involves structural rearrangements within STIM1, allowing it to couple to Orai1 and activate Ca^2+^ entry into the cell.

## 3. Synthetic Biology, Application to CRAC Channels, and Impact on CRAC Channel-Dependent Downstream Signaling

Synthetic biology comprises methods and tools that are used to manipulate basic biological components at the level of DNA, proteins, cells, or even multicellular structures in order to redesign a biological process [207,208,209]. An elegant method of rewiring biological processes is the control of physical distance, which is beneficial for various processes in nature, including cell–cell contacts, signal transduction, and protein transport. Researchers have taken advantage of this fact to better understand cellular signaling processes using small molecules [210], proteases [211], or photosensory modules [212] to induce the proximity of cellular components. Furthermore, in synthetic immunology, engineered chimeric receptors have made it possible to selectively target disease [208]. Arguably, the greatest advantage of induced proximity is the potential to rapidly initiate a downstream signaling process in living cells and monitor its consequences, which enables precise kinetic studies [210,213].

In the following sections, we focus on the application of synthetic biology to CRAC channels and the use of the newly designed CRAC channel tools in downstream signaling cascades relevant in immune cells.

### 3.1. Chemical Inducers of Proximity (CIPs)

Chemical inducers of proximity (CIPs) are small molecules with the special ability to link two proteins, both binding to distinct parts of the dimerizer molecule [214,215,216]. Many of these molecules are naturally occurring; for instance, the macrolactam product, FK506 (Tacrolimus [217]), which binds the prolyl isomerase, FKBP12, on one side and the Ser/Thr phosphatase, calcineurin, on its other side. In nature, the latter is crucial for immunological activation [215]. Additional examples are cyclosporin A, which binds to calcineurin and the prolyl isomerase, CyP, impeding the translocation of the nuclear factor of activated T cells (NFAT) to the nucleus [218], and the immunosuppressant rapamycin, which couples to a prolyl isomerase, FKBP, and a Ser/Thr phosphatase, FRB [219,220] (Figure 1A).

First, evidence for a better understanding of signal transduction through the proximity of signaling molecules has been provided by synthetic small molecule-induced dimerization of the T cell receptor. Indeed, FK1012-induced dimerization of T cell receptors was sufficient to trigger downstream TCR signaling events [210]. Furthermore, the timing of various cellular processes could be better determined by using chemical dimerizers. For instance, transcriptional activity was found to be most efficient when more than 1 signaling input was provided [210,221]. Meanwhile, different variants of chemical inducers of proximity (CIPs) have been used to control protein degradation, induce cell death, drive cell transport mechanisms, regulate gene activation, and impact signal transduction pathways [210].

CIP technologies have already been demonstrated to have the potential for therapeutic approaches, e.g., mitigating complications from engineered immune cells (graft-versus-host disease, B cell aplasia, repopulation after successful transplantation) [210,222,223,224]. In particular, cellular therapies based on the use of CIPs turned out to be promising strategies to guarantee the delivery of precise amounts of therapeutic proteins with temporal control. For instance, selective induction of apoptosis has been achieved through targeted activation of caspase. This and similar strategies are beneficial in gene therapy to accomplish the removal of pathogenic cell types [225,226,227,228,229,230] or to cause the induced apoptosis that is beneficial in preventing graft-versus-host disease [231,232,233,234].

In the CRAC channel field, CIPs have been exploited to study the mechanism of store-dependent Ca^2+^ entry, and in particular, to induce oligomerization of STIM1 and the consequences thereof [184]. The authors substituted the N-terminal Ca^2+^-sensing site of STIM1, which physiologically controls the oligomeric state, with FKBP and FRB proteins. The idea was to keep STIM1 in the resting state as long as no small molecule was added for dimerization. Only by using the rapalog AP21967 they were capable of inducing their oligomerization. This resulted in the ability of these STIM1 proteins to form complexes, localizing to the cell periphery and thus activating Orai1 channels at the cell membrane [184]. This demonstrated that the induced oligomerization of STIM1 was sufficient for Ca^2+^ influx via Orai1 channels (Figure 1B).

The FRB-FKBP dimerization system has been further used to clarify whether STIM1–lipid interactions occur in an oligomerization-dependent manner [188]. To this end, first, FKBP was fused to a nanobody (GNb) against GFP that could bind GFP-STIM1-SOAR. Second, a five-tandem repeat of FRB fused to CFP was used. Only the addition of rapamycin triggered the formation of SOAR clusters, which were absent when the FKBP-GNb construct was missing (Figure 1C). This system confirmed that only the rapamycin-induced oligomerized SOAR state showed binding to phospholipids. To distinguish between binding to PI(4,5)P2 and PI4P, the so-called Pseudojanin (PJ) system [188] was used, which is also based on the FRB-FKRB system. It consists of chimeric phospholipid phosphatase constructs, namely INNP5E and Sac, which exhibit PM recruitment when rapamycin is applied (Figure 1D). INNP5E depletes PIP_2_ through the conversion of PI(4,5)P2 into PI4P, and Sac depletes PI4P through the conversion of PI4P into PI. While the PJ-wild-type system contains both phosphatases, PJ-Sac only depletes PI4P, and PJ-INNP5E only degrades PIP_2_. As a control, another PJ variant is available that does not degrade either phospholipid variant. Using this system, SOAR was shown to bind predominantly to PI4P. This interaction is due to a cluster of conserved lysine regions in SOAR [188]. This approach revealed that STIM1 oligomerization promotes its interaction with the PM-lipids required to bring STIM1 to the ER–PM junctions.

Altogether, CIP technologies have provided new insights into the mechanisms of the CRAC channel machinery. Although they would also have the potential to precisely control downstream signaling pathways via small molecule-triggered dimerization, this has not been exploited so far. A first possible experiment in this direction would be the initiation of NFAT translocation into the nucleus through rapamycin-induced STIM1 oligomerization.

### 3.2. Proteolysis-Based Signaling

A variety of naturally occurring signaling pathways are regulated by proteolysis, which involves either cleavage at a certain site or proteasome-mediated degradation of specific proteins that trigger the exposure of a motif that is critical for degradation. Proteolytically modified effectors can irreversibly initiate signal transduction processes. Meanwhile, proteolytic regulation has been introduced into mammalian cells to control transcription factors and protein interactions [211,235,236,237,238] or even to develop cellular logic by cleaving transcription factors or bound proteins [239,240,241]. Since response rates of transcription factors are naturally slow, the application of proteolysis-based signaling systems to protein interactions and modifications is advantageous, as they occur within the range of minutes [211]. To enable the construction of signaling pathways with proteolytic activity, a toolbox of components has been developed, including orthogonal proteases that act as signal transducers and elements that respond to proteolytic activity and trigger the development of an output signal (Figure 2A). The latter can include binding sites, such as coiled-coil domains, that can be restructured in response to proteolysis [211].

Common proteases with a high orthogonality include tobacco etch virus protease (TEVp) [243,244], plum pox virus protease (PPVs) [244], soybean mosaic virus protease (SbMVp) [245], and sunflower mild mosaic virus protease (SuMMVp) [246]. For dynamic control of their proteolytic activity along the signaling pathway, the proteases have been modified to be inducible by chemical input signals or protein–protein interactions. To achieve this, the proteases have been designed as split proteases. Complementary fragments of the split enzyme have been fused to FKB and FRKB or ABI and PYL1, whose heterodimerization is inducible by rapamycin or abscisic acid, respectively. The addition of the suitable chemical inducer triggered the proteolytic activity [211] (Figure 2B).

Protein–protein interaction modules have been designated as suitable transducer elements responding to proteolysis. Well-understood protein–protein interaction sites include coiled-coil domains, which have been also designed de novo [247,248,249,250]. Canonical coiled-coil regions are distinguished by a repeat of seven residues per two helical turns, each denoted as a heptad repeat with the amino acids assigned “abcdefg” [251]. The “a” and “d” positions represent hydrophobic residues, while the “b”, “e”, and “g” positions are charged residues and the “f” sites represent polar residues [252]. Typically, two or more helices fold into a left-handed supercoiled complex [253]; for instance, for SNARE proteins regulating vesicle fusion or the kinetochores that ensure chromosome segregation [254]. Orthogonal CC pairs possessing different affinities and orientations have been used to develop peptides that act as target, autoinhibitory, or displacer peptides (Figure 2A). To achieve the desired functional output, target, autoinhibitory, and displacer peptides need to be fused to a functional split protein. Based on this principle (platform for the design of a proteolysis-based signaling pathway), a full set of Boolean logic gates could be generated in mammalian cells, which were responsive at a time scale of minutes [211].

To control the STIM1/Orai1 cascade via proteolysis, a genetically engineered protease-activated Orai activator, termed PACE, was used. For this purpose, CAD (STIM1: aa 344-442) fragments were attached to coiled-coil-forming peptide pairs via a protease cleavage site, which locked CAD in an inactive state. The incorporated protease cleavage sites are either those from tobacco etch virus (TEVs) or plum pox virus (PPVs). A defined protease that can be chemically regulated was able to cleave the inhibitory coiled-coil pairs, triggering oligomerization and activation of PACE2 (Figure 2C), as evidenced by Ca^2+^ imaging studies. Jazbec et al. [242] developed a library of PACE constructs. They demonstrated that direct attachment of CAD to the TEV cleavage site and then to the coiled-coil domain leads to spontaneous activation, but it is maintained in the inactive state by the additional insertion of a peptide with a high helical propensity. For efficient and dual regulation, they used not only one type of protease cleavage site for a CAD pair, but also the TEV cleavage site for one CAD and the PPV cleavage site for the second. Maximum activation was only achieved when both were split. They also used spit proteases that can be activated by CIPs (rapamycin, abscisic acid) to trigger cleavage (Figure 2B,C). This tool was valuable to trigger Ca^2+^ entry, NFAT activation, and associated downstream signaling events. The latter was demonstrated by protease cleavage-triggered expression of the cytokine IL-2, which is an essential cytokine governing T cell growth, the differentiation of regulatory T cells, and induction of cell death, as well as TNFα, which is an essential cytokine involved in the immune response, in Jurkat T cells [242]. Despite these valuable effects, further studies are needed to determine how fast PACE can be activated and whether its Ca^2+^ currents correspond to those of CRAC channels.

Overall, proteolytic cleavage is a valuable, albeit irreversible, tool for initiating CRAC channel-dependent downstream signaling processes that are important in immune cells. Nevertheless, the development of strategies for use in immunomodulation in disease is still pending. The published PACE constructs and the PACE variant with two protease sites show that they have the potential to be integrated into larger and more complex circuits. With the ability to switch protease sites, PACE provides a robust framework to link not only synthetic circuits but also any established, well-defined protease function, such as the linkage of cell death-regulating proteases (e.g., caspases, cathepsins) to Ca^2+^ entry, which could be helpful for target specifically controlled immune responses.

### 3.3. Photosensory Domains

As an alternative to chemical- or proteolysis-induced conformational changes, synthetic biology includes optogenetic tools, such as the use of naturally occurring photosensitive proteins. These provide another means for high spatiotemporal control of protein dynamics. Two main groups of photosensitive domains originate in the plant kingdom and are used to transfer their photosensitivity to a protein of interest. They can be divided into those that oligomerize and those that undergo a structural change upon exposure to light (Figure 3A,B). The chromophore, a cofactor bound to these proteins, confers photosensitivity [255].

#### 3.3.1. Light-Induced Oligomerization

Proteins that can oligomerize upon exposure to light include cryptochromes, phytochromes, and UV resistance locus 8 (UVR8). They can be used to remotely and reversibly switch a protein of interest between the inactive and active state. Cryptochrome 2 (AtCRY2), derived from *Arabidopsis thaliana*, contains the flavin adenine dinucleotide as a chromophore and is excited by blue light. The latter leads to electron transfer, which triggers the subsequent flavin reduction and leads to a conformational change in the N-terminal photolyase homology region (PHR) of CRY2. It can rapidly and reversibly homo- or hetero-oligomerize [256,257,258,259] (Figure 3A). The heteromeric complex can be formed with the cryptochrome-interacting basic helix-loop-helix 1 (CIB1) region [260,261,262]. Various factors, such as location in the cell, availability of binding sites, or orientation to each other can determine whether homo- or heteromerization of CRY2 occurs [261]. Phytochromes, unlike cryptochromes, are sensitive to red and far-red light, with the chromophore phytocyanobilin (PCB) conferring photosensitivity. Exposure to red light switches it between the inactive cis- and active trans-isomerization, which triggers interactions with the phytochrome-interacting factor (PIF). Far-red light can reverse this process [263].

**Figure 3 cells-13-00468-f003:**
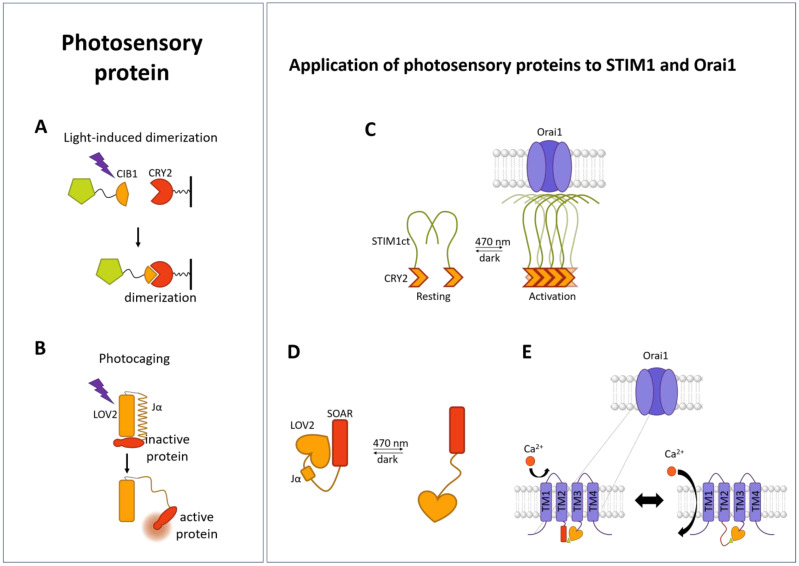
Schematic summarizing the principle of photosensory domains and their application to STIM1 and Orai1. (**A**) Light-induced dimerization of CRY2 (ref) and CIP1 (orange) linked to target protein (green). (**B**) LOV2 domain (orange) linked to a protein of interest (POI (red)) hides the active site of the POI. After UV light irradiation, the linker between LOV2 and the POI undergoes a conformational change and releases its active site. (**C**) CRY2-C-terminal STIM1-fragment (CRY2-STIM1-C-terminus (-Ct)) enables light-triggered homomerization and subsequent coupling to Orai1 to trigger its activation. (**D**) LOV2-STIM1 hides the active site of SOAR in the dark state, while upon irradiation with UV light, it is released. Light-mediated release of SOAR triggers coupling to CC1 under resting cell conditions and coupling to Orai1 in store-depleted cells. (**E**) Light-switchable Orai1 containing the LOV2 domain in the loop2 region allows for light-induced activation of Orai1 (schematics adapted from reference [264]).

In the field of CRAC channels, the optical dimerizer CRY2 has been used to control STIM1 oligomerization, independent of ER store depletion by light. Specifically, the photolyase homology region (PHR-aa 1-498) of the optical dimerizer CRY2 was connected with different STIM1 fragments to induce the activation of STIM1 by light. Mechanistically, blue light stimulation triggers the oligomerization of the respective CRY2-STIM1 fusion protein, while in the dark, the light-sensitive chimera is maintained in the resting state (Figure 3C). This enabled the study of different CRAC channel activation steps, including oligomerization, PM localization, and Orai1 activation [178].

Initially, this concept was implemented by linking STIM1 fragments of different lengths, containing the TM domain and a cytosolic segment, to the AtCRY2 protein, called OptoSTIM1. This enabled light-induced CRAC influx that was activatable within a minute, while deactivation took about 4 min. OptoSTIM1 matches the typical properties of wild-type STIM1, including its movement alongside microtubules, the formation of punctae, localization at ER–PM junctions, and the maintenance of Ca^2+^ selectivity of Orai1 upon stimulation with blue light. Blue light-induced activation of Ca^2+^ entry due to the stimulation of OptoSTIM1 was proven in many cell types, including HEK293, HeLa, HUVECs, NIH3T3, astrocytes, and human embryonic stem cells (hESCs) [265].

Later, two additional light-sensitive STIM1 variants, namely Monster-OptoSTIM1 (monSTIM1) and enhanced OptoSTIM1 (eOS1) [266,267], with improved light sensitivity were developed. monSTIM1 contains the E281A mutation in CRY2, which ameliorates the resting state of the chimera in the dark, and an extension at the C-terminus of CRY2 (A9-ARDPPDLDN), which enhances light-sensitivity. eOS1 is an improved variant of OptoStim1 that contains the CRY2 mutation E490G, which increases oligomerization.

In a different strategy, STIM1 segments were tethered to CIB1, and the provision of soluble CRY2 molecules resulted in the activation of STIM1 fragments upon exposure to blue light.

In addition to the gained high spatial and temporal control over Ca^2+^ signaling, these photosensitive STIM1 variants provided new insights into CRAC channel activation mechanisms. For example, the CRY2-STIM1 tools proved valuable in identifying the individual stages of the STIM1 activation cascade [178]. Thus, the interaction interface of CC1 and SOAR/CAD that forms the inhibitory clamp was resolved by appending STIM1 C-terminal fragments of distinct lengths to CRY2. N-terminal deletions of increasing CC1 lengths demonstrated that the aa250–342 region is crucial for sustaining the inhibitory clamp, in accordance with previous findings. In detail, the CRY2-STIM1 C-terminal chimera was inactive prior to blue light irradiation, while truncation of aa up to L251–L258 in CC1 led to constitutive activation already in the dark [178]. Precise determination of the opposing region in the SOAR/CAD domain has not yet been performed.

CRY2-STIM1 fragments harboring luminal or C-terminal fragments of STIM1 at distinct lengths were also precious for deciphering the segments that are essential for STIM1 oligomerization. Light-induced co-clustering experiments demonstrated that the ER-luminal SAM domain and the cytosolic SOAR region are determinants of STIM1 oligomerization.

Additionally, the identification of new gain- and loss-of-function mutations and the analysis of disease-related mutants was simplified by the utilization of CRY2-STIM1 C-terminus chimeras along with a screening of mutations created by random mutagenesis. Nevertheless, more studies are required to ascertain the cause of the functional alterations in the detected mutants.

In addition, CRY2-STIM1 chimeras facilitated the tracking of the interplay between STIM1 and microtubules as well as the PM. In this regard, the TRIP motif (STIM1_433–640_) [178] in the STIM1 C-terminus was demonstrated to couple specifically with the microtubule plus-end tracking protein EB1, whereas the polybasic cluster at the very end of the STIM1 C-terminus is vital in directing activated STIM1 into ER–PM junctions to instigate STIM1-triggered Orai1 activation [178,268].

#### 3.3.2. Light-Induced Uncaging

Photoresponsive proteins, which undergo a conformational change upon exposure to light, lead to the release of the active site of a protein of interest. They are called light-oxygen-voltage (LOV) domains and are phototropins, a family of blue light-sensing photoreceptors [269]. The LOV domain is composed of a ß-sheet structure, the PAS (Period-ARNT-Singleminded) core, that is bound to the chromophore FMN (flavin mononucleotide), and a C-terminal Jα-helix [270,271]. The LOV2 domain, originating from the phototropin of *Avena sativa* (AsLOV2), is the most popular [272,273]. Upon irradiation with blue light, the Jα-helix is released, which can expose a signal sequence or protein binding interface [272,273] that triggers cellular signaling cascades or enables protein–protein interactions [274,275,276,277] (Figure 3B).

In addition, LOV domains have also been used to trigger protein multimerization. For example, the enhanced light-induced dimer (iLID) containing a LOV2 domain from *Avena sativa* and the Escherichia coli (*E. coli*) peptide SsrA has been used to trigger the oligomerization domains SsrA and SsrB upon exposure to light [278]. To achieve this, SsrA was sterically hindered to bind to SsrB in the dark, but it was released upon activation with blue light. As an alternative to using CRY2, light-triggered oligomerization of STIM1 was attained through heteromerization of the SsrA and SsrB regions. To this end, the enhanced light-induced dimer, iLID, composed of LOV2-SsrA, was employed [278]. While SsrA is hidden in the dark by the *Avena sativa* AsLOV2 domain, it is exposed upon illumination with blue light. Consequently, SsrA can dimerize with SsrB.

Regarding the application of LOV2 in the CRAC channel field, the STIM1 C-terminus was attached to the AsLOV2 photoswitch. Under dark conditions, the Jα-helix strongly couples to the PAS core. This strong interaction assures that STIM1 is kept in the inactive state, while the LOV2 domain masks the active site of the fused STIM1 C-terminal fragment. Exposure to blue light released the tight state of the Jα-helix and the PAS region through conformational changes in the chromophore. Thus, the STIM1 C-terminus is exposed for coupling to Orai1 [279,280] (Figure 3D). Whereas the earliest of these constructs displayed notable dark activity that could not be ignored, truncation of CC1 diminished the dark activity. The reason for the latter is potentially a competition between SOAR for interaction with LOV2 and CC1 domains, hence diminishing the efficiency of caging [280]. The construct Opto-CRAC that exhibited low dark activity contains LOV2_404–546_ and STIM1_336–486_. Its response to light occurs rapidly and reversibly with an activation time of 8 s and a deactivation time of 20 s [280].

A comparison of the kinetics of Opto-STIM1 and Opto-CRAC in different cell types showed that Opto-CRAC leads to more rapid enhancements in Ca^2+^ influx, whereas Opto-STIM1 exhibits a superior quality in driving Ca^2+^ influx. Furthermore, Opto-STIM1 is widely applicable, whereas Opto-CRAC requires the additional coexpression of exogenous Orai1 to enhance Ca^2+^ influx. Under analogue conditions, Opto-CRAC led to an approximately six-fold increase in Ca^2+^ repetitions compared to Opto-STIM1 due to its higher kinetics. Additionally, transient Ca^2+^ signals induced by Opto-CRAC imitated the physiologically induced Ca^2+^ oscillation in mammalian cells.

LOV2-SOAR provided new insights into the mechanisms of the CRAC channel and allowed for the characterization of the essential intramolecular interactions between CC1 and SOAR that sustain the resting state of STIM1. To this end, Ma et al. [178] took advantage of the evidence that STIM1, including the N-terminus, TM, and CC1 domain of STIM1, which is truncated at position 342, couples to soluble SOAR under resting conditions and dissociates upon store depletion. In this way, an elegant assay consisting of STIM1 1-342 and LOV2-SOAR was designed, enabling the assessment of the relative strength of the interactions between SOAR and CC1 in the ER and Orai1 in the PM. In detail, LOV2-SOAR is located in the cytosol, both in cells containing only Orai1 or both Orai1 and STIM1 1-342. Exposure to blue light resulted in the translocation of LOV2-SOAR to the PM because of its coupling to Orai1. Of note, when Orai1 and STIM1 1-342 were expressed at a 1:1 ratio, activated LOV2-SOAR was predominantly located in the ER [178]. This implies that, firstly, SOAR itself interacts preferentially with CC1, and secondly, that further forces are involved to achieve the coupling of SOAR and Orai1 [178]. Different ratios still need to be tested for more accurate characterization of SOAR’s preferences for coupling to CC1 or SOAR. In addition, this method is useful for resolving the CC1-SOAR as well as the SOAR-Orai1 coupling interfaces.

Other LOV2-STIM1 proteins with a similarly high efficiency represent the so-called BACCS variants (blue light-activated Ca^2+^ channel switch). They include hBACCS1, a fusion of LOV2 with the STIM1 C-terminal fragment (aa 347-448), BACCS2, a dimer of BACCS1, and a corresponding *Drosophila melanogaster* form (dmBACCS2), all of which exhibited similar kinetics [281].

In addition to photoresponsive STIM1 proteins, there are also Orai variants, which respond to light. Because STIM1 and Orai1 interact directly, it seemed logical to design a chimeric construct composed of a light-sensitive STIM1 and Orai1 channel. Indeed, Ishii et al. [281] successfully developed fusion proteins of a BACCS variant (hBACCS1, hBACCS2, dmBACCS2) and Orai1 (Orai1:BACCS2). Among the different BACCS forms, the *Drosophila* variant dBACCS2-dOrai exhibited the best activation kinetics. The BACCS variants could be reversibly activated by illumination with blue light and abolished by the removal of extracellular Ca^2+^.

BACCS proteins act very similar to the original CRAC channel activation mechanism, while the use of the LOV2 domain directly on Orai1 also enabled the transfer of light sensitivity. Incorporation of the LOV2 domain into loop2, the flexible loop region between TM2 and TM3, of Orai1 allowed, after further rounds of optimization including N-terminal deletion and single point mutations, for the development of a light-activated Orai1 channel, which is independent of STIM1, termed the light-gated Ca^2+^ channel (LOCa; Orai1 Δ1-64 163AsLOV2164 H171D P245T) (Figure 3E). In detail, the LOCa shows low background activity in the dark and can be reversibly stimulated by blue light [282]. Light-activated LOCa currents display a high Ca^2+^ selectivity and are inhibited by the CRAC channel blocker BTP2; however, these currents are much lower than the currents of overexpressed STIM1/Orai1 or GoF-Orai1 mutants. Additionally, several other properties of CRAC channels, such as rapid Ca^2+^-dependent inactivation, as described in Krizova et al. [205], still need to be characterized.

To date, these photoresponsive Orai forms have been used to modulate and examine cellular downstream processes and disease-related pathways. Still, they have not yet been used to obtain a mechanistic understanding of Orai activation. Notably, LOCa is a powerful tool that has been used to deepen our current comprehension of the role of the loop2 region in Orai1 gating [201,203,283]. Intriguingly, LOV2 integration was able to confer light-mediated Orai1 activation only in the presence of a constitutively active point mutation (P245T). The additional single point mutation (H171Y) suggests that a proper interplay between cytosolic segments is needed, as we recently showed for salt bridge interactions within cytosolic triangles located in Orai1 TM domains [197,198].

#### 3.3.3. Genetically Encoded Light-Sensitive CRAC Channel Components in the Immune Response

Considering downstream signaling involved in immunomodulation, photo-responsive CRAC channel components enabled precise control over downstream events such as gene transcription. Specifically, some of the currently available arsenal of light-sensitive constructs allowed for light-mediated NFAT activation [265,266,279,281,282]. As the frequency of light pulses increased, so did the extent of NFAT translocation into the nucleus [279]. Notably, optogenetic CRAC channel constructs resulted in considerable luciferase/insulin gene expression [279]. CD4+ T cells containing Opto-CRAC produced cytokines, including IL2 and IFN-γ. Human THP-1 macrophages containing Opto-CRAC released IL-1ß and processed caspase-1 after irradiation with light. This supports the role of the opto-CRAC channel in facilitating macrophage-mediated inflammatory responsiveness.

More efficient gene expression was achieved by combining the photoactivatable Ca^2+^ actor, Opto-CRAC, with CRISPR–Cas9 (clustered regularly interspaced short palindromic repeats-associated-9 nuclease) tools [284,285,286]. This strategy allowed researchers to obtain precise and reversible control over the CRISPR–Cas9 system through light-mediated activation of Opto-CRAC and to avoid off-target effects. For this purpose, the so-called Ca^2+^-responsive dCas9 fusion construct (CaRROT), comprising the N-terminal fragment of NFAT (aa 1-460) linked to dCas9 and transcriptional coactivators (VP64/VP160), was produced [287]. In coexpression with the Opto-CRAC construct, the NFAT fragment linked to dCas9-VP64 translocated to the nucleus upon irradiation with blue light. dCas9 in the nucleus was further guided to its target genes by sgRNA (single guide RNA) to trigger gene expression [287].

Incorporation of Opto-CRAC channels into therapeutic dendritic cells in a mouse model of melanoma enabled light-induced Ca^2+^ response in immune cells and fostered the maturation and antigen presentation of dendritic cells. The latter increased priming and activation of T cells, which promoted the decline in melanoma [279].

Bohineust et al. [266] used eOS1 to optically induce Ca^2+^ influx into T cells and influence migration dynamics and chemokine release of CD8+ T cells. Moreover, they used two-photon photoactivation, which allows for deeper tissue penetration, in the popliteal lymph nodes of mice containing eOS1 to mediate light-dependent enhancements in Ca^2+^ levels.

In summary, an array of photoresponsive CRAC channel tools is available to optically and reversibly control cellular functions that are crucial in the Ca^2+^-dependent immune response. They are distinguished by overcoming the restrictions of diverse traditional pharmacological approaches, such as the by-passing of upstream processes or the high speed of light-induced signal transmission in biological systems. Yet, several improvements in their working mechanisms are still necessary to avoid instances of nonspecific interactions. Moreover, a set of factors should be taken into account before their application in the biological processes, including the improvement of illumination conditions to reduce failures within the cellular pathways or the reduction of damage to the biological probe.

### 3.4. Unnatural Amino Acids

Unnatural amino acids represent artificial, chemically synthesized amino acids that do not belong to—but do expand—the pool of naturally occurring canonical amino acids. They possess various novel biophysical or biochemical properties, including light-sensing features, selective reactivity, or posttranslational modifications. Their incorporation into the protein of interest at the desired position has been made possible by genetic code expansion (GCE) technology [288,289,290]. The genetic encoding of UAAs entails the introduction of a specific pair of tRNA and aminoacyl-tRNA synthetase (AARS) into the host cell. These elements are designed to detect the required UAA without interference with the endogenous pairs, thus forming an orthogonal pair. During natural protein translation at the ribosome, the UAA is site-specifically incorporated into the developing protein at the location of a stop codon. This is typically the AMBER (TAG) stop codon, which has been previously inserted into the protein of interest. It has been shown that several UAAs are appropriate for protein incorporation in mammalian cells (Figure 4A).

In the field of ion channels in particular, photoresponsive UAAs have been used to transfer light-sensitivity. This group includes UAAs for optical manipulation (photocaged, photocrosslinking, photoswitchable) and for optical monitoring (fluorescent UAAs) [291,292,293,294,295]. Photocaged UAAs contain a UV light-removable protecting (caging) group [288,296,297,298,299]. Photocrosslinking UAAs (e.g., p-benzophenylalanine (Bpa or BzF), azidophenylalanine (Azi or AzF [288])) remain chemically inert under physiological conditions but transform into highly reactive groups when exposed to UV light (365 nm). They form upon irradiation with UV light (365 nm) covalent bonds (C-H reactions) with nearby backbones and amino acid side chains [292,293,294] in the range of 3–4 Å. They are useful for the identification of protein–protein interaction sites and have already been used to transfer light responsiveness to ion channels or receptors [291,300,301,302]. Light-switchable UAAs toggle between two conformations depending on the applied wavelengths [303,304,305,306,307]. Fluorescent UAAs, such as ANAP or Tyr-Coumarin, sense their environment and thus function as suitable indicators for conformational changes [308,309].

The use of unnatural amino acids in characterizing the function of membrane proteins is currently emerging. Photo- and chemical crosslinking UAAs have been used to study the protein–ligand interactions of G-protein coupled receptors (GPCR) [292,310,311,312]. Photocrosslinking UAAs enabled researchers to study the protein–ligand interactions of a voltage-gated K^+^ ion channel [297,313,314] and to resolve the dynamic functional states of neuronal receptors [307,315]. Functional reversibility of a photoswitchable UAA enabled rapid and reproducible photocontrol of glutamate receptors (NMDAR) [291,307]. A photocaged cysteine was used to generate a light-activatable potassium channel [297]. Among the membrane proteins, fluorescence labelling via UAAs has been achieved for a GPCR [316]. In particular, in the field of TRP channels, fUAAs have been used to resolve structural alterations upon agonist stimulation [309,317,318,319,320,321,322]. Overall, UAAs offer great opportunities to gain highly precise and selective control over protein function and structure [288,301,303].

In the CRAC channel field, we recently demonstrated that the insertion of photocrosslinking UAAs (p-benzophenylalanine, azidophenylalanine) at single positions of Orai1 TM domains allows us to transfer light sensitivity to the entire Orai1 channel complex (Figure 4B). We discovered mutants that showed no or low activity before UV light irradiation upon insertion of a photocrosslinking UAA, which was drastically enhanced after UV light irradiation independent of STIM1. Vice versa, we also identified mutants that showed constitutive activity after incorporation of the UAA, which decreased after exposure to UV light. Detailed characterization of UV light-activatable mutants revealed that they exhibited comparable biophysical properties to typical CRAC channels, indicating that they mimic, at least in part, STIM1-mediated conformational changes [302]. Hence, the insertion of a single photocrosslinking UAA at a certain position in the TM domain revealed that UV-induced local structural rearrangements led to a global conformational change of the entire channel complex, triggering its opening or closing. Furthermore, we showed that UV-induced Orai1 activation is capable of triggering NFAT translocation independently of STIM1 (Figure 4B), highlighting its potential to mediate and modulate the immune response [302].

Photostimulation represents a great venue for researchers to selectively target proteins that lead to pathologic responses. However, as GCE is a multi-component tool, it is still technically demanding to gain precise control of light-sensitive proteins in native tissue. For that, the following steps need to be established: (i) the introduction of the genes of interest, including a protein with the AMBER stop codon and the tRNA/aaRS synthetase pair; (ii) the supply with the respective UAA; and (iii) the efficient delivery of light. In all three aspects, researchers have made some progress; however, a number of hurdles still need to be overcome [291]. The increasing use and further development of UAA incorporation methods will provide new insights into the working mechanisms of proteins and holds promise for the manipulation of processes in physiology and pathophysiology.

### 3.5. Upconversion Nanoparticles

A fundamental hindrance to the application of optogenetic methods in vivo is the incapacity to stimulate deep within tissues without invasive fiber-optic probes. Most of the currently available optogenetic tools respond to UV or visible light. This gives rise to concerns of potential damage due to light stimulation and restricted penetration into tissues, which requires the use of highly invasive implantation of optical fiber devices. Near-infrared (NIR) lanthanide-doped upconversion nanoparticles (UCNPs) have been proven to be suitable for overcoming these disadvantages [323,324,325,326,327,328,329].

UCNPs are luminous donors originating from rare-earth elements, whereas different types have been generated. Depending on their composition, they can emit green or blue light upon excitation with NIR light. In earlier studies (Figure 5A), UCNP stimulation platforms have been engineered to efficiently transfer NIR light to the sample, including, for instance, films with the UCNPs embedded and acting as scaffolds for cell growth or implantable UCNP-packed transducers which convert light from NIR to the visible wavelength range [330,331,332]. To circumvent the constraints of a low penetration of excitatory light and invasiveness of light source implants, UCNPs were incorporated together with genetically expressed light-sensitive ion channels into the cell or organism of interest such as neurons, *Caenorhabditis elegans*, zebrafish, or rats to successfully activate channelrhodopsins [326,333,334,335,336,337,338]. Strategies to further reduce the distance between UCNP and target proteins consist of the specific binding of the UCNPs to the protein of interest, either through streptavidin-conjugated UCNPs or covalent tethering through click-chemistry [323,324,325].

In the Ca^2+^ signaling field, streptavidin-conjugated UCNPs were used as NIR light transducers in close proximity to stimulate an optically sensitive CRAC channel using light in the near-infrared range in deep tissue [279]. The UCNPs used were found to be highly photostable, and their inherent upconversion characteristics (NIR excitation and emission in the visible light range) render them an optimal tool for achieving remote light stimulation of Opto-CRAC channel function [279]. To reach the absorption range of LOV2, UCNPs (40-nm b-NaYF4: Yb, Tm@b-NaYF4 UCNPs) were chosen, which are emitted upon excitation with 980 nm light in the blue region of 470 nm. This was assumed to be suitable to excite recombinant LOV2 proteins with the emitted blue light, which is reversible in the dark state. This shows that a spectral shift towards the NIR region is feasible. To effectively exploit this possibility in living cells, streptavidin-conjugated UCNPs were designed and bound to Orai1 channels, which included a genetically encoded streptavidin-binding tag (StrepTag) in the second extracellular loop (mCh-ORAI1StrepTag). Indeed, it was demonstrated that streptavidin-conjugated UCNPs could be recruited to Orai1StrepTag (Figure 5B) and successfully trigger Ca^2+^ influx via LOV2-Orai1 via NIR light excitation, as detected using various genetically encoded Ca^2+^ indicators [279] (Figure 5C).

This enabled the wireless light-dependent activation of Ca^2+^-dependent signaling processes in a mouse model of melanoma. As a result, antigen-specific immune reactions were stimulated, and tumor growth and metastasis were inhibited upon irradiation using external NIR light [279].

In summary, these are promising approaches for reaching deeper tissue layers. While this has been successfully demonstrated for tissues equipped with light-sensitive proteins, there is still significant potential for their use in combination with unnatural amino acids. Especially for photoswitchable UAAs, which are excitable in the blue light range, these approaches can be well-utilized. Nevertheless, further developments are still needed to introduce these tools into cells and tissues of interest in a targeted manner.

### 3.6. Photosensitive Drugs

Recently, optogenetic attempts have been used to control Ca^2+^ channels with a high spatial and temporal resolution [265,279,339]. However, optogenetic methods are limited in certain cases, as they require the introduction of exogenous genes and the expression of non-native proteins. Conversely, photopharmacological strategies [340,341,342,343] involve the use of photoswitchable molecules to precisely control the effects of bioactive targets such as ion channels [344,345,346] or receptors [347,348,349] in space and time (Figure 6A).

A number of photoswitchable ligands, including photoswitchable-soluble and photoswitchable-tethered ligands, have been designed in the Ca^2+^ ion channel fields. They have not only been used to manipulate the respective channel type but also in native tissue and organisms. This has been previously reviewed [264].

Meanwhile, research on a few photoswitchable ligands that act on CRAC channels has been published. To achieve this, known CRAC channel inhibitors, including GSKs [350,351] and Synta66 [352,353], have been transformed via azobenzene incorporation into various photoswitchable derivatives, piCRACs [344]. Most notably, the so-called piCRAC-1, which is based on the compound GSK-5498A, was capable of reversibly switching CRAC channels on and off via light stimulation at changing wavelengths (Figure 6A,B). In particular, the cis-state of piCRAC-1 was suitable for inhibiting CRAC currents [344]. Recently, Synta66 was reported to bind to a region near the pore [354]. Thus, it is reasonable to assume that the GSKs and piCRAC-1 act at sites near the pore, but further evidence is required for this. Alternatively, another study [355] found that the fusion of the CRAC channel modulator 2-aminoethoxydiphenyl borate (2-APB) with an azobenzene moiety enabled the development of light-switchable CRAC channel modulators called LOCI-1 and LOCI-2 (Figure 6A,B). In particular, LOCI-1 allowed for optical control over CRAC channel function in a highly spatiotemporal manner using alternating irradiation with UV (360 nm) and green (520 nm) light. As it inhibits Orai1, but not Orai3, a chimeric approach was used to show that the LOCI-1 inhibits CRAC channel activity through an extracellular site close to the channel pore [355].

Both compounds, piCRAC-1 and LOCI-1, could be used in vivo in the context of disease [344,355]. piCRAC was shown to allow for optical control of Stormorken syndrome in a zebrafish model to relieve hemorrhage and thrombocytopenia [344]. LOCI-1 was used to gain optical control of Ca^2+^-dependent gene expression in T lymphocytes as well as metastatic cell seeding and nocifensive behavior in mice [355].

Overall, optopharmacology in CRAC channels is suitable to optically interfere with pathological conditions linked to dysbalanced Ca^2+^ signaling. Its combination with UCNPs could further extend its application fields.

### 3.7. Engineered Immune Cells

Engineered antigen receptors have revolutionized the treatment of hematologic malignancies, showing promising outcomes in cell-based therapies combating a variety of cancers. The most clinically employed is the autologous chimeric antigen receptor (CAR) T cell therapy, which has lately entered into the mainstream of certain blood cancer treatments [356]. After isolating T lymphocytes from a patient, cells are genetically modified to express CARs on their surfaces. CARs are synthetic T cell receptors consisting of a tumor-recognizing single chain variable fragment (scFv) fused by a spacer or a transmembrane domain to intracellular T cell signaling domains of the CD3ζ−ITAM subunit [357,358,359] (Figure 7). Upon binding to specific antigen on tumor cells CARs stimulate T cell activation and tumor-killing function [357,358].

T cell priming and activation determine the strength and effectiveness of Ca^2+^ signals via CRAC channels and subsequent immune responses against tumors. Understanding and manipulating Ca^2+^ signaling pathways in T cells can have implications for enhancing the efficacy of CAR-T cell therapy and improving the overall function of T cells [356,357,358,359] (Figure 7).

Although CAR T cell therapy products have been approved by the Food and Drug Administration for clinical applications, they can cause severe side effects such as cytokine release syndrome (CRS) due to uncontrolled release of proinflammatory cytokines and “on-target off-tumor” toxicity. Controlling T cell activities in vivo at a high spatiotemporal resolution can be one solution to this problem. Optogenetics provides versatile tools for the control of cellular functions with superior spatiotemporal precision using light of different wavelengths.

Nano-optogenetic immunotherapy has recently been shown to successfully activate CAR T cells only within the tumor microenvironment and greatly diminish “on-target, off-tumor” cytotoxicity and systematic CRS [360]. The activity of engineered CAR-T cells was precisely controlled by introducing a LOV-based optogenetic device (LOV2-SssrA/sspB) into CARs (OptoCARs). OptoCARs or LiCARs (light-inducible CARs) contain two non-functional parts bearing photo-responsive modules in each part. Part A contains an anti-CD19 single-chain variable fragment (scFv), a costimulatory domain (4-1BB), and an intracellular light-inducible dimerization domain ssrA-cpLOV2, while Part B consists of a T cell receptor-derived CD3ζ subunit and sspB domain as the binding partner of ssrA [360]. After coexpressing the two components encoding constructs in Jurkat cells, the assembly of functional optoCARs upon light stimulation was proven using Ca^2+^- and CRAC channel-dependent NFAT-dependent expression of luciferase and interleukin-2 production readouts for T cell activation. Additionally, photostimulated optoCAR successfully induced cytotoxicity in the co-cultured CD19^+^ Raji tumor cells in an ex vivo model [360]. Furthermore, the efficacy of the opto-CAR T cells was evaluated in mouse models of lymphoma and melanoma. To overcome limited tissue penetration problems, upconversion nanoparticles (UPCNs) or nanoplates were applied as nanotransducers to convert deep tissue-penetrating NIR light into blue light for photostimulation. Biotinylated LiCAR T cells were conjugated to streptavidin-coated UCNPs, and the Stv-UCNP-LiCAR T cells exhibited NIR light-inducible anti-tumor activity, T cell proliferation within the tumor microenvironment, and IFNγ−production in vivo [360] (Figure 7). This nano-optogenetic approach shows spatiotemporal, wireless, and reversible control of CAR T cells, holding the potential for improved precision and safety in cancer immunotherapy [357,358,359,360].

Another class of synthetic antigen receptors that have been engineered is known as synthetic Notch (syNotch) receptors [361,362]. These receptors consist of an antigen binding domain, the Notch core protein derived from the Notch/Delta signaling pathway, and a transcription factor. Instead of initiating T cell signaling upon binding to the target antigen, the Notch protein is enzymatically cleaved by endogenous proteases, thus releasing the transcription factor from the cell membrane. As a result, the subsequent translocation of the transcription factor into the nucleus leads to transcriptional regulation of target genes. However, these engineered immune cells are customized to recognize only one antigen. A linkage between Notch signaling and SOCE was narrowed down from observations in human pulmonary arterial smooth muscle cells, which demonstrated that store-operated Ca^2+^ entry is enhanced upon Notch-activation by the ligand Jeg-1. However, the underlying mechanisms are still unknown, but they could be based on a direct interplay between intracellular Notch proteins and one of the SOCE components [363,364]. Further studies using synNotch could help to clarify this.

To extend the targeting capabilities of CAR and syNotch receptors, a so-called a “universal” receptor system was developed. With this new concept, these receptors were genetically fused with SNAPtag (*O*^6^-methylguanine-DNA methyltransferase), and thereafter, they covalently bound with benzylguanine (BG)-conjugated antibodies. Applying BG antibodies as universal adapters, SNAP-CAR and SNAP-Notch-engineered CAR-T cells enabled researchers to recognize various tumor antigens and consequently reduce tumor size in vivo in a human tumor xenograft mouse model [362].

A promising and complementary approach to optogenetics is to implement designer receptors exclusively activated by designer drugs (DREADDs) for selectively manipulating different cell populations [365]. These chemogenetic tools, heavily applied in neuroscience, show promising outcome for targeting GPCR-mediated downstream Ca^2+^ transduction pathways. There is some evidence that GPCR-mediated Ca^2+^ entry also involves CRAC-dependent signaling pathways leading to subsequent activation of gene transcription programs [366]. However, in-depth investigations would be needed to understand how GPCRs interplay with Orai channels and whether these pathways activate distinct downstream transcription factors in specific subsets of immune cells. Added to this, the employment of photosensitive drugs or optogenetic strategies would contribute to deciphering GPCR-mediated Ca^2+^ signaling pathways with a high spatiotemporal resolution.

To this end, engineered immune cell therapies are now being applied in clinical practice to fight certain blood cancers, but they still suffer from severe side-effects. As a future direction, (nano) optogenetic-engineered immunotherapy promises better and safer control of the immune response in patients and can pave the way for personalized anti-cancer therapy for next-generation precision medicine.

### 3.8. Therapeutic Antibodies, Nanobodies, Antibody Mimetics

Therapeutic antibodies hold a myriad of advantages over small molecules and peptides as a result of their specificity, bioavailability, half-life, and effector functions. However, there are only a few monoclonal antibodies (mAbs) targeting ion channels, which have recently entered into the clinical pipeline. One of these is a novel anti-human Orai1 antibody DS-2741a, which reached clinical phase 1 study in 2020. It is a humanized Fc-silent IgG1 that targets Orai1 for the treatment of atopic dermatitis [367].

Antibody derivatives and mimetics have been developed as promising alternatives to therapeutic antibodies, offering numerous advantages such as their small size, ease of production, high stability, low immunogenicity, and efficient tissue penetration [368,369]. One step further is to spatiotemporally control the activity of antibody derivatives and mimetics over target protein binding using light-controlled and chemically inducible dimerization systems. In recent years, a variety of reversible and light-switchable protein binders have been developed, such as opto-nanobodies (OptoNBs) [370] and OptoMonobodies (OptoMB) fused with the LOV domain [371] as well as Optobodies bearing a Magnet optical dimerization system [372]. Additionally, the reassembly of a chemically switchable split nanobody (Chessbody) using a cpFRP-FKBP-based CID system was demonstrated against different protein targets [373]. In another light-activatable approach, a photocaged variant of the ultra-high affinity ALFA-tag nanobody (ALFA-tag photobody) were prepared using GCE technology [374]. These activatable protein binders can be applied to cells to dynamically control (endogenous) target binding with subcellular spatial precision, thereby modulating signaling pathways. However, their application in exploring cellular downstream processes related to CRAC channels has yet to be explored.

### 3.9. Conclusion and Perspectives

In summary, precise control over the proximity of signaling molecules within the cell offers novel paths to understand protein working mechanisms and their function in physiological and pathophysiological processes. The application of synthetic biology tools to CRAC channels has proven to be suitable for target-specific stimulation of their function and immune cell related downstream signaling cascades. In particular, optogenetic tools enable fast modulation of biological processes within seconds. A key criterion for high spatiotemporal control over protein functions and biological processes is reversible controllability. While several photosensory tools can be controlled in a reversible manner, chemical dimerizers, proteolytic cleavage, or photocrosslinking UAAs can only be operated irreversibly. Among photosensory UAAs, photoswitchable ones have been synthesized and shown to be suitable for reversible manipulation of protein function using two alternating wavelengths [305,306,375]. This, however, has so far only been accomplished for calmodulin [306,375] and glutamate receptors [307]. Nevertheless, such reversibly modulatable optogenetic tools promise to selectively trigger different short- or long-term immune cell responses by altering the oscillation frequency of Ca^2+^ signals through different exposure times. Furthermore, these tools could help to achieve optimal Ca^2+^ levels suitable to achieve cancer killing as well as reductions in cancer cell growth. Alternatively, the combination of distinct synthetic biology tools might be beneficial for target-specific regulation of the various facets of immune responses. For medical applications, deep tissue penetration is mandatory, which among optical tools is only possible using infrared light. This has been successfully achieved via a bypass solution using UCNPs [280], which are excitable in the near-infrared light range and emit light of wavelengths that are suitable for stimulation in the blue/green light range, as shown for photosensory proteins. For the latter application, reversible manipulation between blue light and a dark state was required. For systems to be switched using alternating wavelengths, such as photoswitchable drugs, UCNPs are required, which can be switched between two emission wavelengths through excitation via two wavelengths in the infrared range. Alternatively, a strategy would be to synthesize photosensory compounds that can be switched in the near-infrared range. This, however, requires significant substitutions, typically making molecules bigger and is thus less suitable for modulation of protein function.

In addition to the direct targeting of CRAC channels via synthetic biology tools, Ca^2+^ entry pathways and downstream immune response could be addressed indirectly using chimeric receptors. Currently, cell therapy is performed using the patient’s own cells to avoid problems with immune rejection and is therefore not yet widely applicable. For the latter, approaches would have to be created that make it possible to immunize every cell, which brings safety concerns regarding the safe elimination of these cells. Kill switches offer a new possibility for cell killing, but these are difficult to maintain in the genome due to natural selection. Although engineered immune cells have reached the level of clinical trials, significant research work still needs to be performed in this area to create safe systems for their therapeutic use [208].

Overall, synthetic biology offers great opportunities for the selective modulation of Ca^2+^ signaling processes in immunology and other disciplines. On the one hand, they can provide new insights into the structure/function relationship of the respective modulated proteins, and on the other hand, they are tools for understanding and controlling specific downstream signaling processes.

## Figures and Tables

**Figure 1 cells-13-00468-f001:**
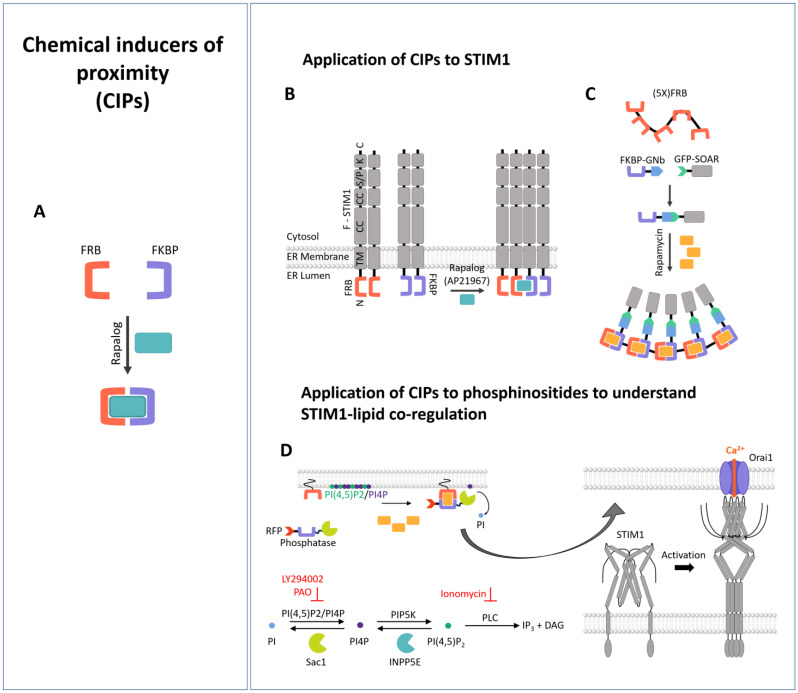
Schematic summarizing the principle of chemical inducers of proximity and their application to STIM1. (**A**) Scheme of the FKBP/FRB system. FKBP (purple) and FRB (red) form a heterodimer in the presence of the rapalog rapamycin (turquoise). (**B**) The FKBP/FRB system has been applied directly to STIM1. Illustration of STIM1 C-termini fused to FRB and FKBP. Their dimerization can be induced by rapalog. (**C**) Alternatively, 5 FRB proteins linked together enable rapamycin-induced SOAR dimerization through interactions with FKBP linked to SOAR (FKBP was fused to a nanobody (GNb (blue)) against GFP (green) that could bind GFP-STIM1-SOAR). (**D**) The FKBP/FRB system has been applied indirectly via PIP-depleting phosphatase to STIM1. Schematic of the methodology for rapamycin-induced phosphatase recruitment to the plasma membrane to induce PIP depletion shows the pathway for PIP_2_ conversion either to IP_3_ and DAG via PLC or to PI4P via INPP5E (blue) and PI by Sac1 (grass green). Rapamycin-induced PIP depletion has been used to understand the role of PIPs in STIM1 activation (schemes adapted from references [184,188]).

**Figure 2 cells-13-00468-f002:**
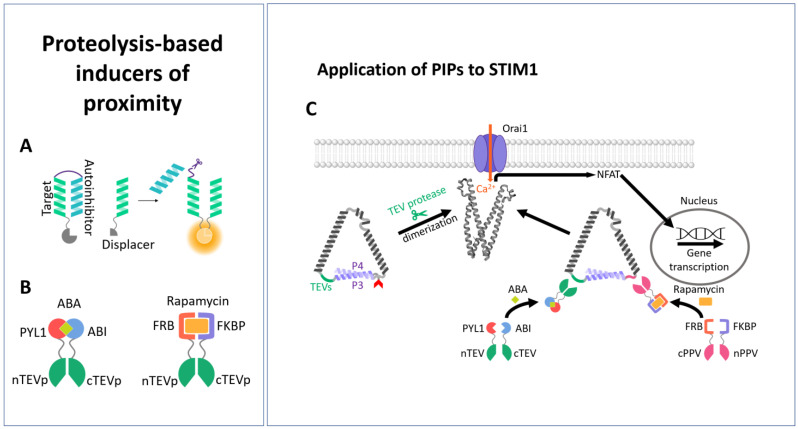
Schematic summarizing the principle of proteolysis-based inducers of proximity and their application to STIM1 and Orai1. (**A**) Proteolytic cleavage-responsive coiled-coil rearrangement composed of a target, autoinhibitory domain, and displacer peptide. After linker cleavage, an autoinhibitory coiled-coil domain is exchanged with a displacer peptide with a higher binding affinity to the target, triggering downstream signaling (as visualized by yellow illuminated circle). (**B**) Schematic displaying the split TEVp reconstituted by ABA and the split PPVp reconstituted by rapamycin. (**C**) Schematic of an engineered STIM1 fragment. The P3 and P4 peptides form coiled-coils to function as inhibitory peptides that impede the self-activation of CAD. The protease cleaves the protease cleavage site between the coiled-coil peptides P3 and/or P4, and CAD and allows for the dimerization of CAD into an active form. As an alternative to adding the protease, split proteases specifically designed for the respective cleavage sites can be used. The protease-activated CAD activates Orai1 and consequently NFAT translocation to the nucleus and gene transcription (schemes adapted from references [211,242]).

**Figure 4 cells-13-00468-f004:**
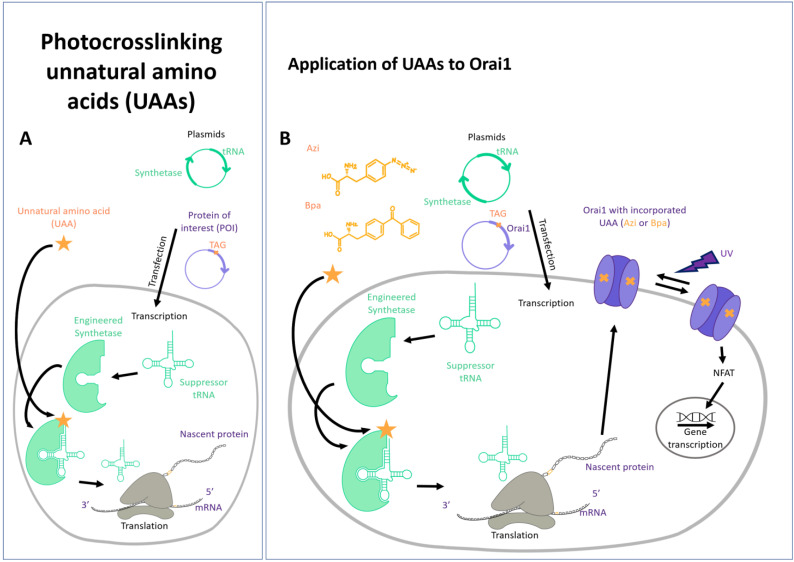
Schematic summarizing the principle of photosensory domains and their application to STIM1 and Orai1. (**A**) CRY2 fused to a target protein allows for light-induced dimerization. (**B**) The LOV2 domain linked to a protein of interest (POI) hides the active site of the POI. After UV light irradiation, the linker between LOV2 and the POI undergoes a conformational change and releases its active site. The CRY2-C-terminal STIM1-fragment (CRY2-STIM1-C-terminus (-Ct)) enables light-triggered homomerization and subsequent coupling to Orai1 to trigger its activation. LOV2-STIM1 hides the active site of SOAR in the dark state, while upon irradiation with UV light, it is released. Light-mediated release of SOAR triggers coupling to CC1 under resting cell conditions and coupling to Orai1 in store-depleted cells. Light-switchable Orai1 containing the LOV2 domain in the loop2 region allows for light-induced activation of Orai1.

**Figure 5 cells-13-00468-f005:**
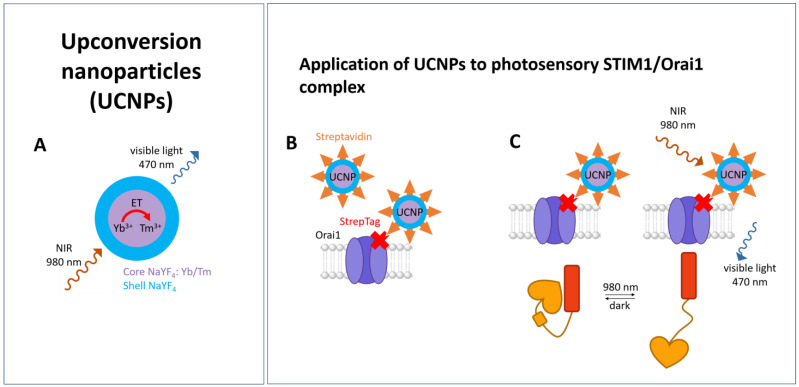
Schematic summarizing the principle of UCNPs and their application to STIM1 and Orai1. (**A**) Schematic of the core structure and energy transfer (ET) among lanthanide ions in NaYF4: Yb, Tm@NaYF4 upconversion nanoparticles (UCNPs). (**B**) Such UCNPs have been specifically targeted via streptavidin conjugation (StrepTag (red cross)) to Orai1 channels in the plasma membrane. (**C**) This allowed for LOV2-STIM1 and subsequent Orai1 stimulation using NIR-light irradiation (schematics adapted from reference [279]).

**Figure 6 cells-13-00468-f006:**
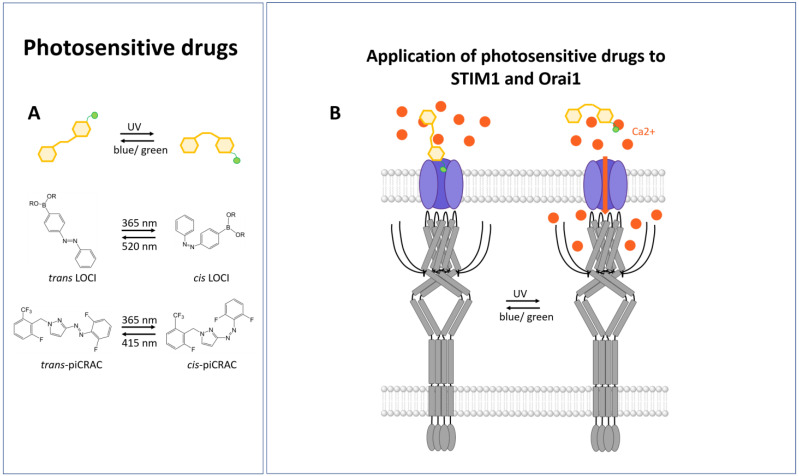
Schematic summarizing the application of photosensitive drugs to the CRAC channel. (**A**) Photosensitive drugs can be switched by the application of two alternating wavelengths between two conformations. Two CRAC channel modulators, LOCI and piCRAC, contain an azobenzene moiety to allow for reversible switching with two wavelengths. (**B**) LOCI and piCRAC have been used to modulate the function of STIM1/Orai1 currents.

**Figure 7 cells-13-00468-f007:**
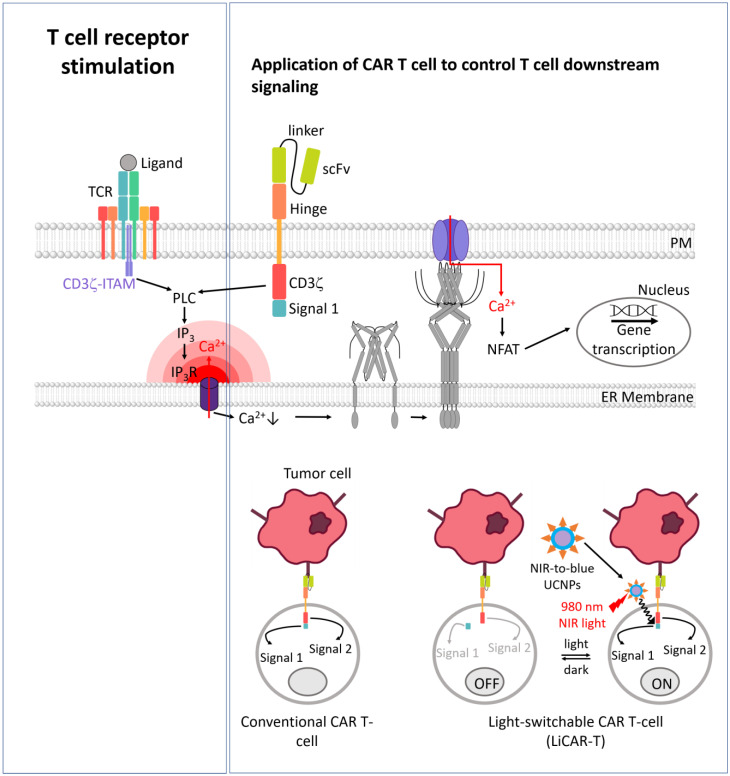
Schematic summarizing the linkage of CRAC channels and the application of CAR-T cells. (**left**) Schematic showing T cell receptors compared to (**right top**) engineered CAR T cell receptors and associated downstream signaling pathways. (**right bottom**) Antigen binding to TCR initiates a PLC-dependent Ca^2+^ signaling pathway to activate customized gene expression. (**bottom**) Schematic summarizing the principle of conventional CAR T cell and UCNP-mediated wireless controllable light-switchable CAR T cells (LiCAR-T). LiCAR-T contains two non-functional polypeptides containing one component of a pair of optical dimerizers in each part (either LOV2-ssrA/sspB or CRY2/CIBN). UCNP converts NIR excitation into blue emission, which subsequently initiates the assembly of optical dimerizers and brings the two CAR parts into close proximity to enable functional CAR reassembly (schematics adapted from reference [360]).

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
