# Peer review of "Synthetic Biology Meets Ca2+ Release-Activated Ca2+ Channel-Dependent Immunomodulation"

_cells, 2024, doi:10.3390/cells13060468_

Round 1

Reviewer 1 Report

Comments and Suggestions for Authors

This review article by Basca et al provides a comprehensive overview of synthetic biology-based methodologies that have been applied to the study of Ca2+ release-activated Ca2+ release channels and the roles of these channels in immunomodulation. The authors do a very nice job of first describing the molecular engineering of each synthetic biology method, and then providing relevant examples from the literature of applications of each method to the study of CRAC channel physiology. The manuscript reflects an incredible amount effort by the authors to read a large number of studies and synthesize them into what will be a valuable resource for the field. I myself learned a lot by reading and reviewing this paper!

I have only a few minor suggested revisions to improve what is already a very well-written paper: 

1)    The authors first use the terminology “store-operated Ca2+ entry (SOCE)” on line 90, but they do not define SOCE or how it relates to CRAC channels. I suggest the authors add a sentence or two explaining the relationship between SOCE and CRAC channels for those less familiar with the field.

2)    On line 130 the authors first refer to “STIM1” and later in the same paragraph to “Orai1”, whereas prior to this they refer simply to “STIM” and “Orai”. I suggest a short explanation of the different STIM and Orai isoforms, or at least a statement that there are 2 STIMs and 3 Orais.

3)    Line 180 defines the “CRAC” abbreviation, but this has already been defined earlier in the manuscript.

4)    On line 199, “5 FKB proteins” should read “FRB”. Similarly, on line 237, change “FKB-FKBP” to “FRB-FKBP”.

5)    I found the paragraph from lines 544 to 555 describing the combination of Opto-CRAC with CRISPR-Cas9 a bit confusing, particularly in terms of what this tool achieves and the function of Cas9 in the construct. I suggest revision of this paragraph for clarity.

Comments on the Quality of English Language

Overall quality of English is good.

Author Response

This review article by Basca et al provides a comprehensive overview of synthetic biology-based methodologies that have been applied to the study of Ca2+ release-activated Ca2+ release channels and the roles of these channels in immunomodulation. The authors do a very nice job of first describing the molecular engineering of each synthetic biology method, and then providing relevant examples from the literature of applications of each method to the study of CRAC channel physiology. The manuscript reflects an incredible amount effort by the authors to read a large number of studies and synthesize them into what will be a valuable resource for the field. I myself learned a lot by reading and reviewing this paper!

We thank the reviewer for the positive evaluation of the manuscript.

I have only a few minor suggested revisions to improve what is already a very well-written paper: 

  • The authors first use the terminology “store-operated Ca2+entry (SOCE)” on line 90, but they do not define SOCE or how it relates to CRAC channels. I suggest the authors add a sentence or two explaining the relationship between SOCE and CRAC channels for those less familiar with the field.

We added a sentence to clarify the relationship between SOCE and CRAC channels at the beginning of section 1.

  • On line 130 the authors first refer to “STIM1” and later in the same paragraph to “Orai1”, whereas prior to this they refer simply to “STIM” and “Orai”. I suggest a short explanation of the different STIM and Orai isoforms, or at least a statement that there are 2 STIMs and 3 Orais.

As requested, we mentioned in section 1 that the different STIM (STIM1, STIM2) and Orai (Orai1, Orai2, Orai3) isoforms play a role in immune cells (page 2, 1st paragraph). Furthermore, we highlighted the complexity of CRAC channels in the first paragraph in section 2.

  • Line 180 defines the “CRAC” abbreviation, but this has already been defined earlier in the manuscript.

Adapted.

  • On line 199, “5 FKB proteins” should read “FRB”. Similarly, on line 237, change “FKB-FKBP” to “FRB-FKBP”.

Adapted accordingly.

5)    I found the paragraph from lines 544 to 555 describing the combination of Opto-CRAC with CRISPR-Cas9 a bit confusing, particularly in terms of what this tool achieves and the function of Cas9 in the construct. I suggest revision of this paragraph for clarity.

We adapted the paragraph to improve clarity.

Reviewer 2 Report

Comments and Suggestions for Authors

The review entitled „Synthetic biology meets Ca2+ release-activated Ca2+-channel dependent immunomodulation” provides an in-depth review of the integration of synthetic biology techniques with the study and manipulation of CRAC channels in immunology. It covers the fundamental roles of CRAC channels in immune response, elaborates on the mechanisms of CRAC channel operation, and highlights how synthetic biology tools are applied to study and potentially modulate these channels for therapeutic purposes.

Based on the sections reviewed, the review is thorough in its explanation of complex biological processes and innovative in its application of synthetic biology to immunology. It also delves into specific strategies like chemical inducers of proximity (CIPs), proteolysis-based signaling, and photosensory domains, showcasing their potential in controlling CRAC channel activity and downstream signaling pathways.

Overall, this review is well written. Section 3 about synthetic biology and its application to CRAC channels was especially exciting to read. The integration of synthetic biology into CRAC channel research opens up vast possibilities for understanding and manipulating immune responses at a molecular level. By controlling the proximity of signaling molecules, inducing specific protein interactions through proteolysis, or utilizing light to modulate CRAC channel activity, researchers can dissect complex signaling pathways and potentially develop novel therapeutic strategies targeting immune-related diseases.

The discussion on photosensory domains is intriguing, especially the part on light-induced oligomerization. However, the section could further benefit from more detailed examples of how these optogenetic tools could be specifically applied to manipulate CRAC channels in immune cells. Any speculation on this?

However, I have some major concerns to raise, particularly in the introduction and the section discussing the CRAC channel working mechanism.

1.    The introduction is quite disappointing, and especially the handling of citations is insufficient. An introduction should provide an overview and a solid and, above all, objective basis for the reader. It should also be provided to readers who are not familiar with the field of calcium channels. It was particularly noticeable that the citations were sometimes very sloppily chosen and did not accurately reflect the intended statement. This leads to a one-sided view of the rather complicated field of calcium signaling. Such oversight undermines the article's credibility and fails to properly contextualize the study within the broader scientific conversation.

Furthermore, the apparently one-sided nature of the citations raises concerns about the breadth and depth of the literature review conducted for this publication. It is crucial for the integrity and credibility of scholarly work to present a balanced view that includes diverse perspectives and acknowledges the contributions of all researchers involved in advancing knowledge in the field. This selective referencing detracts from the nuanced understanding necessary for comprehending such a complex biological process, leaving readers without a clear or complete picture of the current knowledge landscape. Such shortcomings in citation practices diminish this section's immediate value and potential contribution to ongoing scientific discourse.

The reliance on secondary sources such as review articles, while useful for providing an overview, often does not give due credit to the pioneering researchers and the original studies where these findings were first reported. This practice not only diminishes the recognition of the original authors' contributions but also potentially biases the reader's understanding of the topic by not presenting the foundational research directly.

Additionally, in this context, not only is there an excessive reliance on review articles, but there is also a noticeable issue with several citations being duplicated. For instance, citations 19 and 36, 21 and 29, as well as 38 and 41 and so on, are identical. This repetition suggests that the citations were selected not only randomly but also carelessly.

The issue with citations requires attention throughout the entire article.

Line 113-114: The mention of 'recent review articles' is misleading, as only two out of the four cited articles are recent. Please remove 'recent' from the text for accuracy.

In lines 55-58, the citations provided do not accurately support the content or the intended message of this sentence.

The article would also benefit from briefly mentioning other conditions, such as autoimmune diseases and immunosenescence associated with aging, where CRAC channels play a significant role or are altered, if applicable.

2.    Even more concerning was the treatment of the CRAC channel working mechanism section, where the citations were partly wrong but also largely one-sided, favoring a narrow selection of sources. It is particularly crucial to elucidate for the reader the complexity of the interaction between STIM and Orai and the challenges it poses to researchers. The authors state, 'The CRAC channel forms a complex of STIM and Orai isoforms (16, 36, 52, 53)'” (line 127), yet they fail to adequately highlight or detail this complexity. Moreover, the citations provided do not accurately reflect the STIM isoforms, such as STIM2 with its splice variants, nor do they mention the Orai2 and Orai3 channels. The section overlooks the diversity of Orai and STIM isoforms and their specific roles in different tissues or under different physiological conditions. Addressing this could provide a more nuanced understanding of CRAC channel functionality. This oversight might be significant, especially when considering the importance of precisely manipulating and controlling this interaction for therapeutic purposes.

Minor points:

The subsections “Therapeutic antibodies,…” and “Conclusion and perspectives” have the same section number (lines 849 and 872)

Please check the text for misspellings and missing abbreviations

Author Response

The review entitled „Synthetic biology meets Ca2+ release-activated Ca2+-channel dependent immunomodulation” provides an in-depth review of the integration of synthetic biology techniques with the study and manipulation of CRAC channels in immunology. It covers the fundamental roles of CRAC channels in immune response, elaborates on the mechanisms of CRAC channel operation, and highlights how synthetic biology tools are applied to study and potentially modulate these channels for therapeutic purposes.

Based on the sections reviewed, the review is thorough in its explanation of complex biological processes and innovative in its application of synthetic biology to immunology. It also delves into specific strategies like chemical inducers of proximity (CIPs), proteolysis-based signaling, and photosensory domains, showcasing their potential in controlling CRAC channel activity and downstream signaling pathways.

Overall, this review is well written. Section 3 about synthetic biology and its application to CRAC channels was especially exciting to read. The integration of synthetic biology into CRAC channel research opens up vast possibilities for understanding and manipulating immune responses at a molecular level. By controlling the proximity of signaling molecules, inducing specific protein interactions through proteolysis, or utilizing light to modulate CRAC channel activity, researchers can dissect complex signaling pathways and potentially develop novel therapeutic strategies targeting immune-related diseases.

The discussion on photosensory domains is intriguing, especially the part on light-induced oligomerization. However, the section could further benefit from more detailed examples of how these optogenetic tools could be specifically applied to manipulate CRAC channels in immune cells. Any speculation on this?

We thank the reviewer for the positive evaluation of the synthetic biology sections.

Concerning examples of how optogenetic tools can be used to manipulate CRAC channel function in immune cells, we spent a separate section in 3.3.3.

However, I have some major concerns to raise, particularly in the introduction and the section discussing the CRAC channel working mechanism.

  1. The introduction is quite disappointing, and especially the handling of citations is insufficient. An introduction should provide an overview and a solid and, above all, objective basis for the reader. It should also be provided to readers who are not familiar with the field of calcium channels. It was particularly noticeable that the citations were sometimes very sloppily chosen and did not accurately reflect the intended statement. This leads to a one-sided view of the rather complicated field of calcium signaling. Such oversight undermines the article's credibility and fails to properly contextualize the study within the broader scientific conversation.

Furthermore, the apparently one-sided nature of the citations raises concerns about the breadth and depth of the literature review conducted for this publication. It is crucial for the integrity and credibility of scholarly work to present a balanced view that includes diverse perspectives and acknowledges the contributions of all researchers involved in advancing knowledge in the field. This selective referencing detracts from the nuanced understanding necessary for comprehending such a complex biological process, leaving readers without a clear or complete picture of the current knowledge landscape. Such shortcomings in citation practices diminish this section's immediate value and potential contribution to ongoing scientific discourse.

The reliance on secondary sources such as review articles, while useful for providing an overview, often does not give due credit to the pioneering researchers and the original studies where these findings were first reported. This practice not only diminishes the recognition of the original authors' contributions but also potentially biases the reader's understanding of the topic by not presenting the foundational research directly.

Additionally, in this context, not only is there an excessive reliance on review articles, but there is also a noticeable issue with several citations being duplicated. For instance, citations 19 and 36, 21 and 29, as well as 38 and 41 and so on, are identical. This repetition suggests that the citations were selected not only randomly but also carelessly.

We apologize that we have only chosen a small selection of sources. We have checked all references in detail, added not only review articles, but also the direct sources, where applicable, and ensured that we give a broad overview of the research field. Nevertheless, we have kept this section rather general, as it is sufficient to understand the applications of synthetic biology tools to CRAC in immunology. Furthermore, duplicated references (a problem which occurred due to troubles with the referencing program) were removed.

The issue with citations requires attention throughout the entire article.

Line 113-114: The mention of 'recent review articles' is misleading, as only two out of the four cited articles are recent. Please remove 'recent' from the text for accuracy.

We removed the two old review articles and added a series of newer ones, which appeared in the last five years.

In lines 55-58, the citations provided do not accurately support the content or the intended message of this sentence.

This is corrected to better support the intended message.

The article would also benefit from briefly mentioning other conditions, such as autoimmune diseases and immunosenescence associated with aging, where CRAC channels play a significant role or are altered, if applicable.

We added brief information on the aspects autoimmune disease and immunosenescence associated with aging where CRAC channels play a significant role. Nevertheless, we decided not to go into detail on these aspects, as it is not required for the understanding of the synthetic biology part.

  1. Even more concerning was the treatment of the CRAC channel working mechanism section, where the citations were partly wrong but also largely one-sided, favoring a narrow selection of sources. It is particularly crucial to elucidate for the reader the complexity of the interaction between STIM and Orai and the challenges it poses to researchers. The authors state, 'The CRAC channel forms a complex of STIM and Orai isoforms (16, 36, 52, 53)'” (line 127), yet they fail to adequately highlight or detail this complexity. Moreover, the citations provided do not accurately reflect the STIM isoforms, such as STIM2 with its splice variants, nor do they mention the Orai2 and Orai3 channels. The section overlooks the diversity of Orai and STIM isoforms and their specific roles in different tissues or under different physiological conditions. Addressing this could provide a more nuanced understanding of CRAC channel functionality. This oversight might be significant, especially when considering the importance of precisely manipulating and controlling this interaction for therapeutic purposes.

We apologize that we have only chosen a small selection of sources. We have checked all references in detail, added the correct ones and ensured that we give a broad overview of the research field.

As synthetic biology tools have not yet been applied to other STIM and Orai isoforms, the differences in the properties and mechanisms of the different isoforms are beyond the scope of this review. Nevertheless, we mentioned in section 1 that the different STIM (STIM1, STIM2) and Orai (Orai1, Orai2, Orai3) isoforms play a role in immune cells. Furthermore, we highlighted the complexity of CRAC channels in the first paragraph in section 2.

Minor points:

The subsections “Therapeutic antibodies,…” and “Conclusion and perspectives” have the same section number (lines 849 and 872)

Thank you for spotting the mistake. It’s adapted.

Please check the text for misspellings and missing abbreviations

Done.